

# Environmental and spatial drivers of taxonomic, functional, and phylogenetic characteristics of bat communities in human-modified landscapes

Laura M. Cisneros[1], Matthew E. Fagan[2] and Michael R. Willig[3,4]

[1] Department of Natural Resources and the Environment, University of Connecticut, Storrs, CT, United States
[2] Department of Geography and Environmental Systems, University of Maryland Baltimore County, Baltimore, MD, United States
[3] Department of Ecology and Evolutionary Biology, University of Connecticut, Storrs, CT, United States
[4] Center for Environmental Science and Engineering, University of Connecticut, Storrs, CT, United States

Corresponding author
Laura M. Cisneros,
laura.cisneros@uconn.edu

## ABSTRACT

**Background**. Assembly of species into communities following human disturbance (e.g., deforestation, fragmentation) may be governed by spatial (e.g., dispersal) or environmental (e.g., niche partitioning) mechanisms. Variation partitioning has been used to broadly disentangle spatial and environmental mechanisms, and approaches utilizing functional and phylogenetic characteristics of communities have been implemented to determine the relative importance of particular environmental (or niche-based) mechanisms. Nonetheless, few studies have integrated these quantitative approaches to comprehensively assess the relative importance of particular structuring processes.
**Methods**. We employed a novel variation partitioning approach to evaluate the relative importance of particular spatial and environmental drivers of taxonomic, functional, and phylogenetic aspects of bat communities in a human-modified landscape in Costa Rica. Specifically, we estimated the amount of variation in species composition (taxonomic structure) and in two aspects of functional and phylogenetic structure (i.e., composition and dispersion) along a forest loss and fragmentation gradient that are uniquely explained by landscape characteristics (i.e., environment) or space to assess the importance of competing mechanisms.
**Results**. The unique effects of space on taxonomic, functional and phylogenetic structure were consistently small. In contrast, landscape characteristics (i.e., environment) played an appreciable role in structuring bat communities. Spatially-structured landscape characteristics explained 84% of the variation in functional or phylogenetic dispersion, and the unique effects of landscape characteristics significantly explained 14% of the variation in species composition. Furthermore, variation in bat community structure was primarily due to differences in dispersion of species within functional or phylogenetic space along the gradient, rather than due to differences in functional or phylogenetic composition.
**Discussion**. Variation among bat communities was related to environmental mechanisms, especially niche-based (i.e., environmental) processes, rather than spatial mechanisms. High variation in functional or phylogenetic dispersion, as opposed to functional or phylogenetic composition, suggests that loss or gain of niche space is driving the progressive loss or gain of species with particular traits from communities along the human-modified gradient. Thus, environmental characteristics associated with

landscape structure influence functional or phylogenetic aspects of bat communities by effectively altering the ways in which species partition niche space.

## INTRODUCTION

An ongoing quest in ecology is to understand the relative importance of mechanisms that drive community assembly or disassembly. Variation in community structure (e.g., species composition) may arise due to niche-based processes (e.g., environmental filtering, niche partitioning) in which interspecific differences in ecological, morphological or physiological traits dictate species distributions and abundances along environmental gradients (*Weiher & Keddy, 1995*). Alternatively, variation in community structure may arise as a consequence of species-specific spatial dynamics, such as dispersal limitations (*Legendre, 1993*). Nevertheless, environmental characteristics associated with niche-based processes are often spatially structured so that environmental control on community structure may result in patterns that are similar to those produced by spatial processes (i.e., induced spatial dependence; *Legendre, 1993*). Disentangling the confounded effects of environmental and spatial processes on community structure would significantly advance the conceptual underpinning of community ecology.

### Environment versus space

A number of studies have addressed how to decouple the effects of space and environment. Indeed, variation partitioning (*Borcard, Legendre & Drapeau, 1992*; *Borcard & Legendre, 1994*) has become a routine procedure to differentiate among environmental and spatial mechanisms that structure communities (e.g., *Legendre & Legendre, 1998* and sources therein, *Cottenie & De Meester, 2004*; *Leibold et al., 2004*; *Cottenie, 2005*; *Legendre, Borcard & Peres-Neto, 2005*; *Peres-Neto et al., 2006* and sources therein; *Stevens, López-González & Presley, 2007*; *Meynard et al., 2013*). This approach provides a means to decompose variation in community structure into a proportion that is uniquely explained by environmental characteristics ([a] in Fig. 1) and a proportion that is uniquely explained by spatial characteristics ([c] in Fig. 1), by removing the environmental variation that is spatially structured ([b] in Fig. 1). Although findings from this approach provide broad understanding of the relative roles of environment and space in community assembly, they provide little insight on the importance of particular environmental or niche-based processes (e.g., environmental filtering, niche partitioning, or interspecific competition).

Approaches incorporating functional or phylogenetic characteristics of communities facilitate an assessment of the importance of particular environmental processes. Functional and phylogenetic aspects of communities use ecological traits and evolutionary histories of taxa, respectively, to differentially weight the presence of species. Also, phylogenetic patterns often can be interpreted with regard to ecological traits, as many traits exhibit

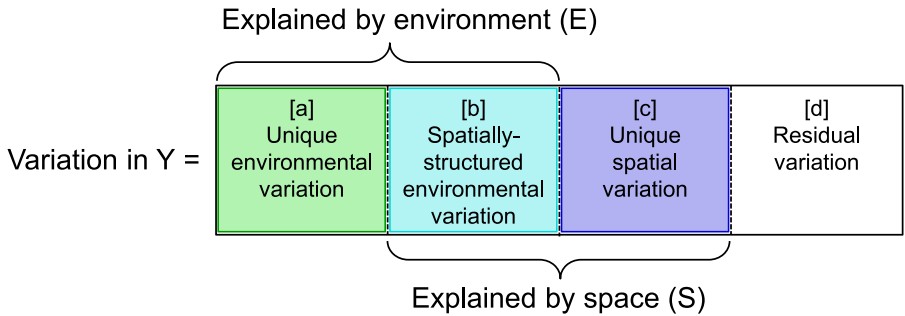

**Figure 1** **Graphic illustrating variation partitioning of a response matrix or vector (Y) with respect to two predictor matrices related to environment (E) and space (S).** Total variation (Y) is partitioned into unique variation explained by E ([a] = [abc] − [bc]), the fraction of variation explained jointly by E and S ([b] = [abc] − [a] − [c]), unique variation explained by S ([c] = [abc] − [ab]), and variation not explained by either E or S ([d] = 1 − [abc]). Total variation (Y) explained by different fractions is expressed in the following notation (after *Legendre, 1993*): [abc], environmental and spatial variation together; [ab], environmental variation; [bc], spatial variation; [a], unique environmental variation after accounting for space; [b], spatially-structured environmental variation; [c], unique spatial variation after accounting for environment; and [d], residual variation.

strong phylogenetic signals (i.e., tendency of closely related species to have more similar ecological traits than expected by chance; *Revell, Harmon & Collar, 2008*). Because the effects of environmental variation are mediated by species characteristics (e.g., physiological constraints, habitat requirements, and dispersal abilities), integration of assessments of functional and phylogenetic patterns into approaches that broadly differentiate the influences of environmental or spatial factors can provide resolution on competing mechanisms.

Functional or phylogenetic structure are each characterized by two general components: mean location (hereafter composition) and dispersion. Composition characterizes the central position of a community within functional or phylogenetic space based on the averages of species characteristics (Fig. 2A), and is conceptually similar to community-weighted means (*Peres-Neto, Leibold & Dray, 2012*). Dispersion measures the distribution of all species in a community with respect to functional or phylogenetic characteristics (Fig. 2B), and is conceptually similar to a variety of metrics that measure functional diversity, such as Rao's quadratic entropy (*Botta-Dukát, 2005*). Environmental or spatial factors can cause variation in functional or phylogenetic structure via (1) shifts in composition of communities with little effect on species dispersion (Fig. 2C); (2) differences in the dispersion of species with little effect on composition (Fig. 2D); or (3) changes in both composition and dispersion.

Until recently, a single method that decomposes total functional or phylogenetic structure into the two components had not been developed (*Peres-Neto, Leibold & Dray, 2012*). By determining the relative importance of functional or phylogenetic components, we can better identify structuring mechanisms associated with environmental factors. For example, significant variation in functional or phylogenetic composition accompanied by little variation in functional or phylogenetic dispersion suggests the operation of environmental filtering, in which communities comprise species that have particular characteristics that are obligatory for persistence at that part of the gradient (*Weiher & Keddy, 1995*;

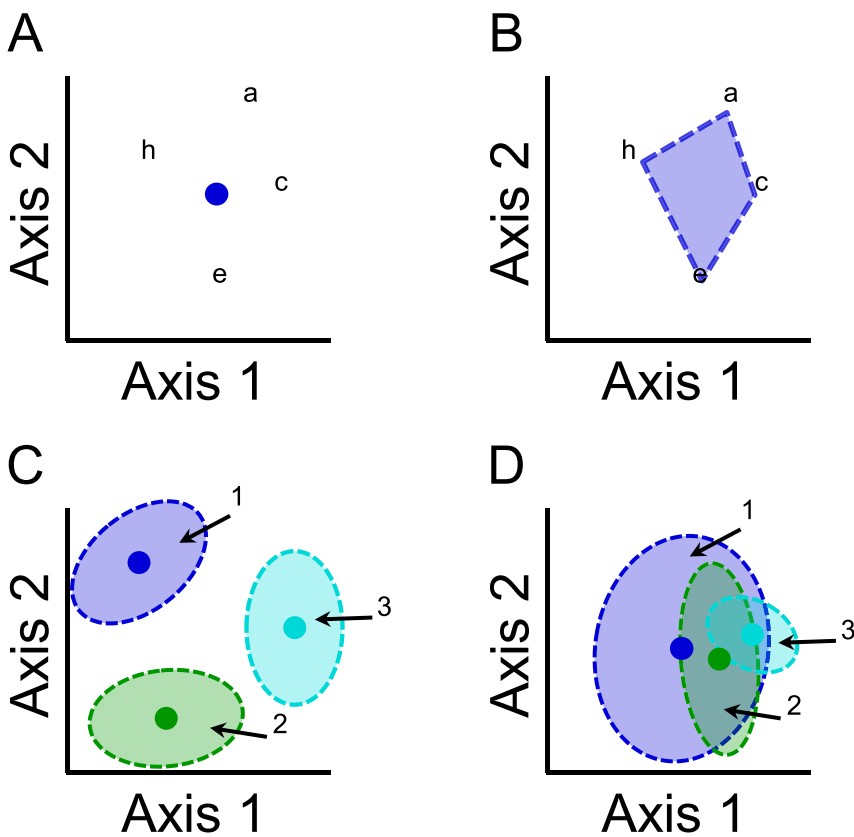

**Figure 2** **Illustrations of (A) the mean location (composition) component and (B) the dispersion component of functional or phylogenetic structure of a community that comprises species a, c, e, and h.** Species are mapped onto functional or phylogenetic space, and distances between pairs of species quantify differences in species characteristics. (A) The composition component can be represented by the centriod (blue dot) of the distribution of species. (B) The dispersion component can be represented by the volume of space occupied by the community (blue shaded region). Illustrations demonstrating that variation in functional or phylogenetic structure of three communities (community 1, blue; community 2, green; community 3, turquoise) can arise from (C) differences in composition or (D) differences in dispersion.

*Mayfield & Levine, 2010*). In contrast, significant variation in functional or phylogenetic dispersion accompanied by little variation in functional or phylogenetic composition suggests the operation of mechanisms associated with variation in niche partitioning (*MacArthur & Levins, 1967*; *Mayfield & Levine, 2010*). That is communities may include more species with unique traits as more niches become available. The relative importance of such niche-based mechanisms is dependent on the role of environmental factors versus spatial factors.

To date, only one study has integrated variation partitioning and phylogenetic approaches to comprehensively assess the relative importance of particular mechanisms (*Gavilanez & Stevens, 2013*). At a regional scale, they found that taxonomic structure and phylogenetic structure of Neotropical primate communities were more influenced by spatial attributes than by environmental or historical factors. Moreover, they found that partitioning of phylogenetic structure revealed complex interactions among environmental, historical, and spatial processes.

## Landscape variation and bats

Little is known about how landscape structure affects variation in functional or phylogenetic aspects of communities (*Tscharntke et al., 2012*). Because fragmentation *per se* is a mesoscale phenomenon when associated with human land-conversion (i.e., scales between local and regional), processes that operate at mesoscales (e.g., environmental heterogeneity, landscape connectivity, dispersal limitation; *Leibold, 2011*) are likely to be influenced by both environmental and spatial factors. As such, disentangling the effects of landscape characteristics and space on variation in functional or phylogenetic attributes of communities can advance the understanding of assembly or disassembly processes in human-modified landscapes.

Bats are useful for assessing the effects of human-modified landscapes on functional and phylogenetic structure because they are diverse from taxonomic, evolutionary, and ecological perspectives (*Patterson, Willig & Stevens, 2003*). In the Neotropics, bats are generally the most species-rich and locally abundant mammalian group, comprise species from a variety of feeding guilds (e.g., frugivores, gleaning animalivores), and vary greatly in dispersal abilities (*Patterson, Willig & Stevens, 2003*). Moreover, bats provide important ecological functions, such as seed dispersal, pollination, and regulation of insect populations (*Kunz et al., 2011*). Due to their diversity and ecological importance in many tropical ecosystems, bats may be keystone taxa as well as bioindicators of disturbance, as their responses to environmental variation may reflect the responses of other taxa (*Jones et al., 2009*).

We employed a novel variation partitioning approach to comprehensively evaluate the relative importance of particular environmental and spatial processes affecting taxonomic, functional, and phylogenetic structure of bat communities within a human-modified landscape. We estimated the unique and shared effects of landscape characteristics (i.e., environment) and space on two components of functional and phylogenetic structure (i.e., composition and dispersion) as well as on taxonomic structure (i.e., species composition). To provide further insights on the importance of particular niche-based mechanisms, we examined the extent to which variation in functional and phylogenetic structure was due to composition or dispersion. Because bats are highly vagile, and variation in a variety of ecological and evolutionary aspects of bat communities is influenced by landscape characteristics (*Gorresen et al., 2005*; *Klingbeil & Willig, 2009*; *García-Morales, Badano & Moreno, 2013*; *Cisneros, Fagan & Willig, 2015a*; *Cisneros, Fagan & Willig, 2015b*; *Meyer, Struebig & Willig, 2015*), we predict that unique effects of landscape structure account for more variation in taxonomic, functional, and phylogenetic structure than do unique effects of space. Nevertheless, we expect the effects of landscape characteristics to be more pronounced on functional and phylogenetic structure than on taxonomic structure based on the assumption that particular landscape characteristics select for species with particular traits.

# MATERIALS & METHODS

## Study area and sites

Research was conducted in a human-modified landscape of the Caribbean lowlands in northeastern Costa Rica (Fig. 3). The 160,000 hectare landscape encompasses fragments of

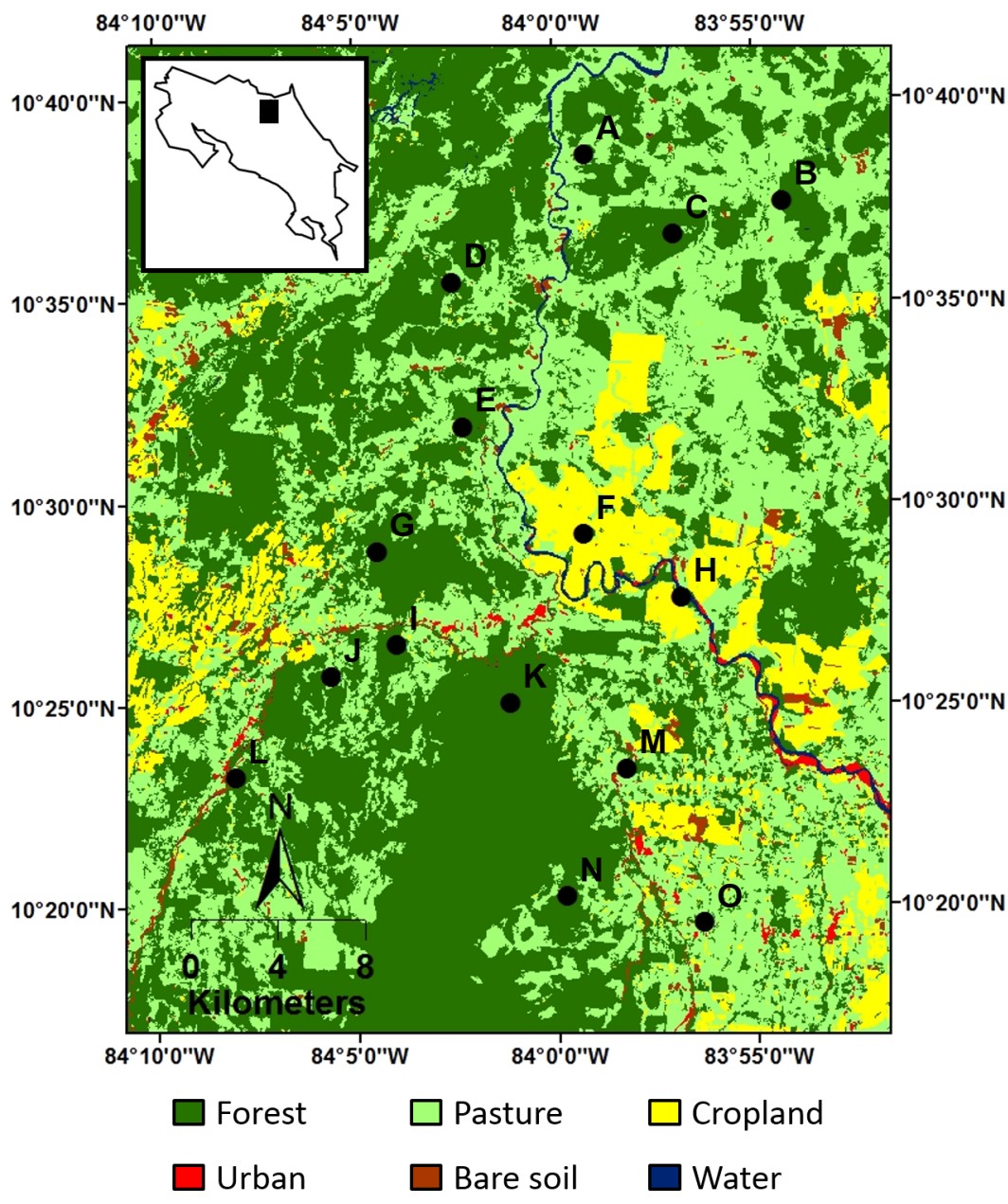

**Figure 3 Location of the 15 sampling sites (black dots) within the study landscape, as represented by a 2011 land cover map.** Location of the study landscape (black rectangle) in Costa Rica is displayed in the inset map of the land cover map. Alphabetical site codes correspond with those in Fig. 5.

wet tropical forest at various successional stages (e.g., old-growth and secondary forests), a variety of agricultural plantations (e.g., heart of palm, banana, and pineapple), cattle pastures, and logged areas. The climate is warm and moist, with relatively constant temperatures throughout the year (mean daily temperature: 31.0 °C; range: 30.2–31.9 °C) and appreciable rainfall every month (mean annual precipitation: 4374.6 mm; range: 2809.3–6164.0 mm; *Organization for Tropical Studies, 2012*). In general, a drier period occurs from January to late April, with mean monthly rainfall of 223.7 mm, followed by a wet period

from early May to December, with mean monthly rainfall of 435.0 mm. Nevertheless, the dry and wet seasons during this study (2010) were less distinct (i.e., mean monthly rainfall was 353.8 mm in the dry season and 431.4 mm in the wet season). Because of changes in resource availability and resource requirements of bats between seasons (*Frankie, Baker & Opler, 1974*; *Tschapka, 2004*), analyses were conducted separately for the dry and wet seasons.

Fifteen circular sites (5 km radius) were established across the landscape so that centers were positioned within forest patches and were separated by at least 3.5 km (Fig. 3). These sites were selected to represent a forest loss and fragmentation gradient that encompasses the current range in composition and configuration of forest land cover in the study area (Table S1). Site selection was not stratified or randomized because of limitations associated with gaining permission from owners to access land and because landscape structures were not equally accessible.

## Data
### *Biological surveys*
Bats were surveyed using ground-level mist nets during the dry season (January to April) and wet season (May to September) of 2010. These surveys provided information on species composition and abundances that were used to quantify taxonomic, functional and phylogenetic structure. Each site was surveyed four times each season. For each survey, 12 mist nets (12 m × 2.5 m) were opened for six hours from dusk until midnight (mist nets were inspected every 30 min). Mist nets were deployed in trails and flyways in closed-canopy forest within 1 km of the center of each site. Sampling was not conducted during the presence of a moon that was ≥90% full due to reduced bat activity associated with high lunar illumination (*Morrison, 1978*) or during severe weather because of health risks to bats from exposure to low temperatures associated with strong winds or rain. To identify recaptures within a sampling period, hair was trimmed on the back of each bat before it was released. The use of ground-level mist nets, unaccompanied by other sampling methods, effectively samples species from the family Phyllostomidae (*Kalko, 1997*), but may under-represent species in other families (*Kalko & Handley, 2001*). Accordingly, analyses were limited to phyllostomids. The protocol for animal use for this research was approved by the IACUC at the University of Connecticut (IACUC number: A09-014). A field research permit was issued by the Sistema Nacional de Áreas de Conservación del Ministerio del Ambiente y Energia of Costa Rica (permit number: 003-2010-SINAC).

### *Functional and phylogenetic characteristics*
Functional characteristics were based on 27 categorical (binary) and mensural attributes that represented seven niche axes (Table S2). Use of multiple niche axes and multiple attributes within niche axes is critical to comprehensively represent functions of species in the ecosystem. Categorical niche axes were associated with (1) diet, (2) foraging location, (3) foraging strategy, and (4) roost type. Mensural niche axes were associated with (1) body size, (2) masticatory mode (i.e., skull characteristics), and (3) aerodynamic mode (i.e., wing characteristics). For each categorical attribute, a species received a ''1'' if it exhibited the characteristic or a ''0'' if it did not exhibit the characteristic. For each mensural attribute,
an average value was obtained for each species based on measurements of multiple male and female adults ($\geq 2$ individuals).

Information for functional attributes was derived from the literature and restricted to records from Central America when possible (Table S3). Additionally, measurements of size attributes were augmented by field measurements from the study area. Missing mensural data were estimated using the least squares regression line between mass and the particular attribute using known values of other species from the same subfamily. Missing categorical data were replaced by values from congeners. Only 4.1% of species traits were estimated or replaced (i.e., 38 of 918 traits).

Functional differences between species were obtained using the Gower metric (R package "clusters"; *Maechler et al., 2012*) and were represented in a pairwise distance matrix. The Gower metric can quantify dissimilarities when simultaneously considering categorical and mensural attributes (*Botta-Dukát, 2005*). Using this metric, functional dissimilarity between species is the sum of the weighted functional trait differences. When all niche axes were used to derive the distance matrix, we weighted attributes such that each niche axis had equal influence on overall structure despite having unequal numbers of attributes. Equal weights were assigned because we have little *a priori* knowledge of which functional niche axes are most important to community assembly. This weighting approach also decreases sensitivity of the analyses to potentially redundant attributes. We also conducted analyses for each functional niche axis separately (Table S2) because the environmental gradient may affect functional niche axes differently (see *Cisneros, Fagan & Willig, 2015a*). Thus, integrating all ecological attributes into a single multivariate measure may obscure important patterns (*Spasojevic & Suding, 2012*).

Phylogenetic characteristics were based on branch lengths from a species-level supertree of bats (*Jones, Bininda-Emonds & Gittleman, 2005*). Five of the 34 species were not present in this supertree. The closest congener present in the supertree that was not present in the study area was substituted for each missing species. Although a number of phylogenetic trees are available for bats, the supertree developed by *Jones, Bininda-Emonds & Gittleman (2005)* represents the most complete and accepted tree. Moreover, higher-level divergences in the selected supertree are consistent with those in other phylogenetic trees (*Jones, Bininda-Emonds & Gittleman, 2005*), and these cladistics events have the predominant effect on phylogenetic characteristics of community structure. Phylogenetic distances between species were calculated via the "cophenetic" function of the R package "ape" (*Paradis, Claude & Strimmer, 2004*) and represented in a pairwise dissimilarity matrix.

### Environmental characteristics

Landscape characteristics were quantified at each site from a land cover map that represented the landscape of 2011 (see *Fagan et al., 2013* for a detailed description of map construction). The original 13 cover types were reclassified into six cover types: forest (i.e., mature forest, swamp forest, native reforestation, and exotic tree plantations), cropland (i.e., banana, sugarcane, heart of palm, and pineapple), pasture, bare soil, urban, and water. Pixel values that were originally designated as masked areas (i.e., areas obscured by cloud or Landsat 7 line errors; 0.6% of land cover) were manually changed to other pixel

values within which they were embedded or to pixel values based on a 2005 land cover map of the study area (*Fagan et al., 2013*) using the area fill tool in ERDAS IMAGINE 2013.

Five compositional (i.e., percent forest, percent pasture, mean forest patch size, forest patch density, and Simpson's diversity of cover types; Table S4) and four configurational (i.e., mean forest proximity, mean forest nearest neighbor, mean forest patch shape, and forest edge density; Table S4) landscape indices were quantified using FRAGSTATS version 4 (*McGarigal, Cushman & Ene, 2012*). Composition refers to the proportions of different types of land cover within a site, whereas configuration reflects the geometric arrangement of land cover within a site. All indices were quantified using forest as the focal land cover type, except for percent pasture and Simpson's diversity of cover types. Spatial patterns are scale dependent, and the scale at which bats use and respond to the environment is species-specific (*Gorresen et al., 2005*; *Klingbeil & Willig, 2009*). Thus, all landscape characteristics were quantified at each of three spatial scales (circles of 1, 3, and 5 km radius) to account for interspecific differences in bat home range size and behavior, as well as to facilitate comparison with other studies on landscape ecology on Neotropical bats.

### Spatial characteristics

Spatial predictors were estimated from Moran's eigenvector maps (MEMs; *Dray, Legendre & Peres-Neto, 2006*). MEMs provide a more powerful means to describe spatial effects at a variety of scales, can explain more variation in species data than can geographic coordinates or polynomial functions of geographic coordinates, and better control for type I error rates associated with unique environmental effects (*Legendre, Borcard & Peres-Neto, 2005*; *Peres-Neto & Legendre, 2010*). To derive these eigenvectors, we first use geographic coordinates of the sites to create a distance matrix. From the distance matrix, a connectivity matrix is constructed based on a threshold distance and minimum spanning tree algorithm. Finally, eigenvectors were computed from the centered connectivity matrix. A single eigenvector associated with a large and positive eigenvalue was used as the spatial predictor because it represents positive spatial autocorrelation and a landscape-wide spatial trend (however, the six MEMs representing different spatial scales all yielded similar results). Construction of MEMs was completed using algorithms written in Matlab by *Peres-Neto, Leibold & Dray (2012)*.

## Statistical analysis

Variation partitioning was conducted for taxonomic, functional, and phylogenetic structure for each combination of season (i.e., dry and wet) and scale (i.e., 1, 3, 5 km). First, structure for each dimension was quantified using a site-by-species abundance matrix as follows:

1. Taxonomic structure—a matrix representing variation in species composition was obtained by employing a Hellinger transformation on the site-by-species abundance matrix. This transformation gives low weights to species that are rare at particular sites and infrequent at most sites (*Legendre & Gallagher, 2001*). Dispersion cannot be quantified for taxonomic structure because all species are considered equally different from one another (i.e., nominal data); thus, there is no variation in interspecific differences.

2. Functional and phylogenetic structure—vectors representing variation in functional (or phylogenetic) composition and dispersion were quantified using the methodologies

of *Peres-Neto, Leibold & Dray (2012)*. First, total functional (or phylogenetic) variation was quantified by linking the site-by-species abundance matrix with an eigenvector representing functional (or phylogenetic) variation among species (eigenvector was derived from a functional or phylogenetic distance matrix) via the Hadamard element-wise multiplier. The total functional (or phylogenetic) variation matrix was weighted based on the sum of the occurrences of each species to minimize the effects of rare species. Next, the total functional (or phylogenetic) variation matrix was decomposed into the composition component ($\bar{X}_F$ or $\bar{X}_P$) and the dispersion component ($s_F$ or $s_P$) by re-distributing the sum-of-squares of the total variation matrix in terms of means and variances among sites.

Next, full and partial redundancy analyses (*Borcard, Legendre & Drapeau, 1992*) were used to partition variation in taxonomic structure (species composition; $\bar{X}_T$), functional (or phylogenetic) composition ($\bar{X}_F$ or $\bar{X}_P$), or functional (or phylogenetic) dispersion ($s_F$ or $s_P$) into variation explained by environmental (i.e., landscape characteristics) and spatial (i.e., MEM) predictors. Three weighted least-squares regressions quantifying (1) variation explained by both sets of predictors (E and S in Fig. 1), (2) variation explained by only the environmental predictors (E in Fig. 1), and (3) variation explained by only the spatial predictor (S in Fig. 1) were used to subsequently partition total variation into four fractions (i.e., unique environmental effects after accounting for space [a], spatially-structured environmental effects [b], unique spatial effects after accounting for environment [c], and residual variation [d]; Fig. 1). In each regression, adjusted $R^2$ was quantified to minimize the bias associated with the number of independent variables and sample size (*Peres-Neto et al., 2006*).

Two permutation procedures were used to test for statistical significance of the unique contributions of environment and space (i.e., fractions [a] and [c], respectively). For taxonomic structure, 1,000 permutations of the community matrix were conducted. For functional or phylogenetic structure, we employed a procedure developed by *Peres-Neto, Leibold & Dray (2012)* that permutes site vectors in the predictor matrix (E or S) 1,000 times and permutes species vectors in the functional or phylogenetic eigenvector 1,000 times. Variation partitioning of taxonomic structure, and associated permutation procedures, were conducted using the function varpart from the R package "vegan" (*Oksanen et al., 2009*). Variation partitioning of functional or phylogenetic structure, and associated permutation procedures, were conducted with algorithms written in Matlab by *Peres-Neto, Leibold & Dray (2012)*.

## RESULTS

Along the forest loss and fragmentation gradient, bat communities exhibited significant variation in structure with regard to species richness (sites ranged from 9–19 species in dry season and 6–20 species in wet season) and abundance (sites ranged from 30–272 individuals in dry season and 16–202 individuals in wet season). In addition, variation in ecological and evolutionary aspects of community structure characterized the gradient, but was primarily due to dispersion ($s_F$ and $s_P$) rather than composition ($\bar{X}_F$ and $\bar{X}_P$; Table S5).

That is, the average functional or phylogenetic characteristic of each bat community (i.e., composition; $\bar{X}_F$ and $\bar{X}_P$) was essentially the same for all sites, but the breath of functional or phylogenetic characteristics (i.e., dispersion; $s_F$ and $s_P$) exhibited by each community differed among sites. Because little variation in functional or phylogenetic composition characterized bat communities within the study area, only results from analyses using dispersion as the response variable (Figs. 4B and 4C, Tables S6 and S7; but see Table S8 for variation partitioning results of $\bar{X}_F$ and $\bar{X}_P$), in addition to species composition, are discussed hereafter.

The total variation in species composition (i.e., taxonomic structure) of bat communities that was explained by both environmental and spatial predictors ranged from 15.0 to 39.4% in the dry season and from 0 to 15.3% in the wet season (see Tables S6 and S7). Most of the variation in species composition was explained by the unique effects of landscape characteristics ([a] in Fig. 4A); however, these significant effects were limited to the dry season at 3 and 5 km focal scales.

In contrast, the total variation in functional or phylogenetic dispersion that was explained by both environmental and spatial predictors ranged from 90.6 to 99.9%, regardless of season or scale (Tables S6 and S7). Unique effects of landscape characteristics ([a] in Figs. 4B, 4C, Table S7) and unique effects of space ([c] in Figs. 4B, 4C, Table S7) on variation in functional or phylogenetic dispersion were relatively small compared to effects of spatially-structured landscape characteristics ([b] in Figs. 4B, 4C, Table S7). On average, spatially-structured landscape characteristics accounted for ~84% of the variation in functional or phylogenetic dispersion, whereas unique effects of landscape characteristics and unique effects of space accounted for ~12% and ~1% of variation in functional or phylogenetic dispersion, respectively. Although the unique effects of landscape characteristics and the unique effects of space on functional or phylogenetic dispersion were small, a few were significant at 3 and 5 km scales during the dry season (Fig. 4B, Table S7).

## DISCUSSION

This is one of the first studies to employ a novel variation partitioning approach for multiple dimensions of community structure, including the decomposition of functional and phylogenetic structure into composition and dispersion components. Through this approach, we comprehensively evaluated the relative importance of various environmental and spatial processes driving community assembly or disassembly within a human-modified landscape.

### Relative roles of environmental and spatial mechanisms

Forest loss and fragmentation due to human land use is a mesoscale phenomenon (i.e., between local and regional scales). As such, a number of environmental or spatial processes may influence communities (e.g., environmental heterogeneity, landscape connectivity, dispersal limitation; Leibold, 2011). Indeed both significant effects of landscape structure and significant effects of space on community structure have been reported at the meso-scale. For tropical dry forests, both unique effects of landscape structure and unique effects of space played significant roles in governing species density (Hernández-Stefanoni et al., 2011). Conversely, the amount of different land cover types, as opposed to space and

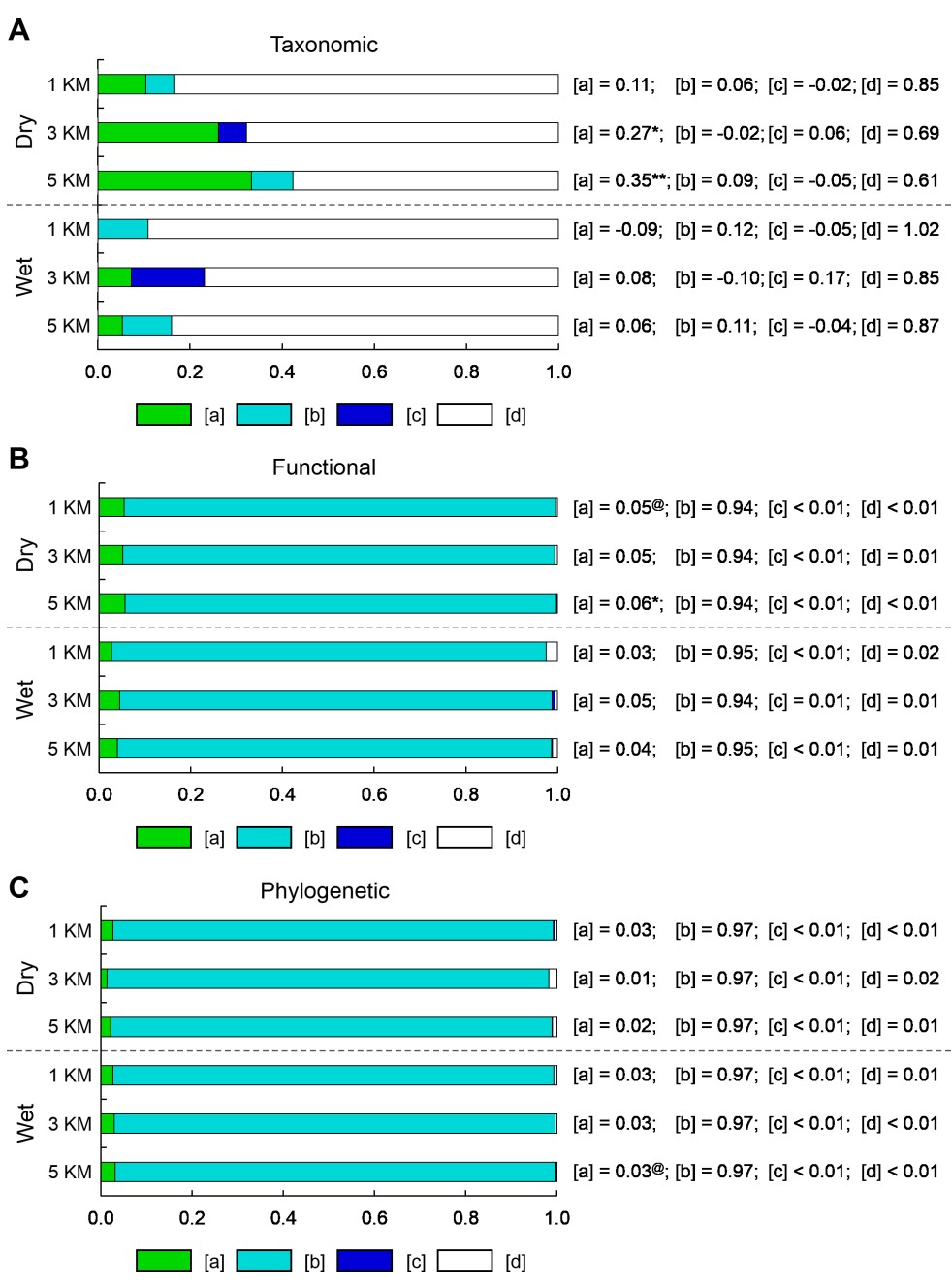

**Figure 4   Bar graphs represent the results of variation partitioning of (A) taxonomic structure based on species composition, (B) functional dispersion based on all functional attributes, and (C) phylogenetic dispersion for each combination of season (wet and dry) and scale (1, 3, and 5 km).** The adjusted percentages of unique environmental effects [a], spatially-structured environmental effects [b], unique spatial effects [c], and residual variation [d] are reported next to each bar. Adjusted $R^2$ can be negative for any fraction and these are interpreted as zeros (*Legendre, 2008*). Negative [b] fractions can occur when explanatory variables are correlated but have strong and opposite effects on the response variable, or when explanatory variables have a weak correlation with the response variable but strong correlation with other explanatory variables that are correlated with the response variable (*Peres-Neto et al., 2006*). Significant testable model fractions (i.e., unique effects) are indicated by superscript symbols (@, $0.10 \geq P > 0.05$; *, $0.05 \geq P > 0.01$; **, $P \leq 0.01$).

configurational characteristics of the landscape, had unique effects on species composition of avian communities (*Heikkinen et al., 2004*). In this study, landscape characteristics played a more appreciable role in structuring Neotropical bat communities, whereas effects that were uniquely attributed to space were consistently small (Figs. 4B, 4C, Tables S6 and S7). The strong role of environmental mechanisms, more so than spatial mechanisms, during community assembly in human-modified landscapes is likely a common theme for mobile taxa that are not as limited by dispersal.

From the perspective of all three dimensions, landscape characteristics were important in molding bat communities. However, landscape characteristics that were not confounded with space dictated species composition (i.e., taxonomic structure; Fig. 4A), whereas spatially-structured landscape characteristics dictated the breath of functional or phylogenetic characteristics (i.e., dispersion) of communities (Figs. 4B, 4C, Table S7). First, the influences of spatially-structure landscape characteristics were not due to space *per se*. This is supported by a lack of spatial autocorrelation in functional or phylogenetic structure of bat communities (Table S9). Second, the difference in landscape effects suggests that processes structuring different aspects of bat communities operate at different scales. Landscape characteristics within 5 km of centers of sites were not spatially autocorrelated (Table S10), but landscape patterns at scales larger than 5 km around sites were inherently spatially auto-correlated due to the nature of land use in the study area (e.g., clustering of certain land cover types; Fig. 3). Thus, broad landscape patterns (>5 km around sites) influenced community assembly via species ecological or evolutionary characteristics, whereas smaller landscape patterns (≤5 km around sites) influenced the species composition of bat communities.

The strength of landscape effects differed between seasons, with more prominent effects during the dry season. Season-specific responses to human land conversion and land use have been observed in other bat communities in Latin America (*Willig et al., 2007*; *Klingbeil & Willig, 2010*). The increased importance of landscape structure to Caribbean lowland bat communities in the dry season is likely driven by a decrease in resource quantity and diversity during this time of year (*Frankie, Baker & Opler, 1974*; *Tschapka, 2004*). In another study, we found that bat species concentrate activities at sites with particular landscape structures that were associated with the presence of diverse food resources during times of limitation, whereas when food resources were more plentiful, bat species were not restricted to sites with particular landscape characteristics (*Cisneros, Fagan & Willig, 2015b*).

During the dry season, unique effects of space had small but significant influences on the breath of diet and morphological traits of bat communities (Table S7). These effects could represent the influences of unmeasured environmental characteristics or the influences of dispersal limitations, which may restrict bats with particular morphological traits or diet requirements to particular sites. Dispersal may become especially important during the dry season as species may need to use a greater number of resources patches to meet dietary demands (*Sih, 2011*).

## Effects on dispersion versus composition

Landscape and spatial predictors explained different amounts of variation in each dimension. Both sets of predictors explained relatively little variation in species composition

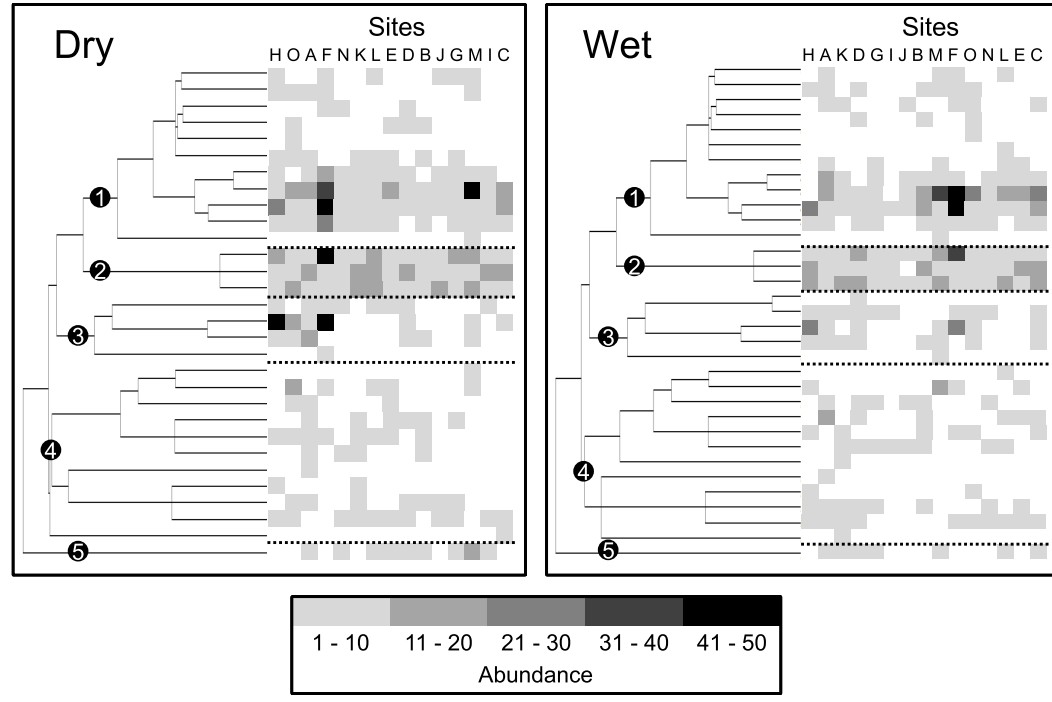

**Figure 5** **Dispersion of species abundances (represented by the intensity of gray) on the supertree for bats at 15 sites within a human-modified landscape during the dry season and the wet season, separately.** Each column (positioned to the right of the supertree) represents a site. Sites are organized from left to right based on decreasing degree of dispersion. Alphabetical site codes correspond with those in Fig. 3. Clades representing five bat subfamilies are indicated by numbered black dots (Stenodermatinae, 1; Carolliinae, 2; Glossophaginae, 3; Phyllostominae, 4; Desmodontinae, 5). Dashed lines separate the five subfamilies to illustrate differences in the representation of subfamilies along the gradient.

(i.e., taxonomic structure), but explained most of the variation in functional or phylogenetic dispersion (Table S6). This difference is due to disparate effects of landscape structure on composition and dispersion aspects of bat communities in the Caribbean lowlands. Species (i.e., taxonomic), functional and phylogenetic composition did not differ greatly among communities because a subset of species from the genera *Artibeus* and *Carollia* (i.e., genera within clades 1 and 2 in Fig. 5) were present in moderate to high abundances at all sites. Given that these abundant species have similar functional characteristics (e.g., species primarily consume fruit) and positions in the bat phylogeny, little inter-site variation characterized average taxonomic, functional and phylogenetic structure throughout the study landscape. Accordingly, composition did not capture variation in bat community structure, and varied little with regard to the environmental and spatial gradient.

Variation in the breath of functional and phylogenetic characteristics (i.e., dispersion) of communities primarily described changes in bat community structure. This variation in structure arose as some sites only had the most abundant frugivorous species from the genera *Artibeus* and *Carollia*, where others comprised additional species that were less abundant from the subfamilies Glossophaginae, Phyllostominae, and Desmodontinae (clades 3, 4, and 5 in Fig. 5). These rarer species possess different functional characteristics from those species in the genera *Artibeus* and *Carollia* (e.g., species consume nectar or pollen, invertebrates

or vertebrates and blood, respectively); thus, communities comprising these species were characterized with a greater diversity of functional and phylogenetic characteristics. As such, processes driving community assembly or disassembly of Neotropical bats in human-modified landscapes likely operate by creating or destroying environmental features that support these non-frugivorous species.

## Relative roles of specific environmental mechanisms

Decomposition of total functional and phylogenetic structure into composition and dispersion provides valuable ecological insights into niche-based (i.e., environmental) mechanisms driving community assembly or disassembly, especially the relative roles of environmental filtering and niche partitioning. Environmental filtering is a process by which species with particular characteristics are unable to persist or establish at a part of the environmental gradient because they are not well adapted or are poor competitors (*Weiher & Keddy, 1995*; *Mayfield & Levine, 2010*). A large influence of environmental filtering in community assembly would manifest through significant shifts in functional or phylogenetic composition. Niche partitioning mechanisms (e.g., interspecific competition, variation in productivity, creation or loss of habitats) reduce or increase the number of potential niches available to species (*MacArthur & Levins, 1967*; *Mayfield & Levine, 2010*). A significant role of niche partition mechanisms in community assembly would impact the breath or dispersion of functional or phylogenetic characteristics of communities, such as in the case of this study.

Human land use and land conversion in the Caribbean lowlands impacts bat communities by modifying niche availability for less common species from the subfamilies Glossophaginae, Phyllostominae, and Desmodontinae. One possible niche partitioning mechanisms relevant to human-modified landscapes is restricted niche availability due to habitat loss or landscape homogenization (e.g., landscapes dominated by monocultures; *Devictor et al., 2008*; *Flynn et al., 2009*). This mechanism is particularly relevant to phyllostomid bats, as most species generally avoid sun-grown monocultures, and species of the subfamily Phyllostominae are generally dependent on complex vegetation structure (*García-Morales, Badano & Moreno, 2013*).

Another potential niche partitioning mechanism relevant to phyllostomid bats arises from the creation of a diversity of new habitats (i.e., increasing landscape heterogeneity due to only moderate amounts of human land use). Many resources used by phyllostomids can be obtained from human-modified environments (i.e., forest edge, pasture and non-monoculture agricultural systems; *Wilkinson, 1985*; *Lobova et al., 2003*; *Thies & Kalko, 2004*; *Harvey & González Villalobos, 2007*). In fact, nectarivores and sanguinivores were more common at sites with a mix of agriculture, pasture and forest than at sites dominated by forest (Fig. 5), as these taxa feed on flowers and fruits from early successional plants and crops (*Lobova et al., 2003*; *Harvey & González Villalobos, 2007*) and on blood from cattle (*Wilkinson, 1985*), respectively. Indeed, areas with intermediate amounts of forest and pasture harbored higher levels of functional and phylogenetic bat diversity in the Caribbean lowlands during the dry season (*Cisneros, Fagan & Willig, 2015a*). Accordingly, increasing niche availability that accompanies increasing landscape heterogeneity is likely the primary driver of community assembly of phyllostomid bats in the Caribbean lowlands.

## CONCLUSIONS

By linking taxonomic, functional and phylogenetic community structure to approaches based on variation partitioning facilitates a more comprehensive assessment of environmental and spatial mechanisms that drive community assembly. Community structure of Neotropical bats in a human-modified landscape was primarily molded by large-scale environmental variation associated with landscape structure, rather than by spatial processes. Landscape structure influenced bat communities via increasing or decreasing niche availability along the human-modified landscape gradient.

## ACKNOWLEDGEMENTS

Special thanks to R Chazdon, D Civco, B Klingbeil, S Presley and M Urban for guidance in the analytical component of this research, to B. Rodríguez Herrera and A Sanchun for guidance in the field, and to R Urbina, H. Lara Perez, K. Díaz Hernández and S. Padilla Alvarez for assistance with field work. We also thank P Peres-Neto for helpful discussions on methodologies, and W. Pineda Lizano and C Meyer for contributions of unpublished measurements of wing characteristics. Logistical support in Costa Rica was provided by Tirimbina Biological Reserve, La Selva Biological Station, Fundación para el Desarrollo de la Cordillera Volcánica Central (FUNDECOR), Ministerio de Ambiente (MINAE), the Wildlife Refuge Nogal, Selva Verde Lodge and Rainforest Reserve, Hacienda Pozo Azul, and two local landowners.

### Funding

This research was supported by a Student Research Scholarship from Bat Conservation International, a Research Fellowship from the Organization for Tropical Studies, two Grants-in-Aid Awards from the American Society of Mammalogists, and many intramural awards from the Center for Environmental Sciences and Engineering, Department of Ecology and Evolutionary Biology, and Center for Conservation and Biodiversity, all at the University of Connecticut (UCONN). Especially noteworthy, field work, data analysis, and manuscript preparation were supported by a Multicultural Fellowship from the Graduate School at UCONN. Furthermore, funding for the synthetic portion of this project was provided by a National Science Foundation grant to S. Andelman and J. Parrish entitled "The Dimensions of Biodiversity Distributed Graduate Seminar" (DEB-1050680). The funders had no role in study design, data collection and analysis, decision to publish, or preparation of the manuscript.

### Grant Disclosures

The following grant information was disclosed by the authors:
Bat Conservation International.
Organization for Tropical Studies.
American Society of Mammalogists.

Center for Environmental Sciences and Engineering.
Department of Ecology and Evolutionary Biology.
Center for Conservation and Biodiversity.
UCONN Graduate School.
National Science Foundation: DEB-1050680.

## Competing Interests

The authors declare there are no competing interests.

## Author Contributions

- Laura M. Cisneros conceived and designed the experiments, performed the experiments, analyzed the data, contributed reagents/materials/analysis tools, wrote the paper, prepared figures and/or tables, reviewed drafts of the paper.
- Matthew E. Fagan contributed reagents/materials/analysis tools, reviewed drafts of the paper, development of land cover map.
- Michael R. Willig conceived and designed the experiments, wrote the paper, reviewed drafts of the paper.

## Animal Ethics

The following information was supplied relating to ethical approvals (i.e., approving body and any reference numbers):

The protocol for animal use of this research was approved by the IACUC at the University of Connecticut (IACUC number: A09-014).

## Field Study Permissions

The following information was supplied relating to field study approvals (i.e., approving body and any reference numbers):

A field research permit was issued by the Sistema Nacional de Áreas de Conservación del Ministerio del Ambiente y Energia (permit number: 003-2010-SINAC).

## Data Availability

Cisneros LM, Fagan ME, Willig MR (2014) Data from: Season-specific and guild-specific effects of anthropogenic landscape modification on metacommunity structure of tropical bats. Dryad Digital Repository: http://dx.doi.org/10.5061/dryad.9fp3g.

## Supplemental Information

Supplemental information for this article can be found online at http://dx.doi.org/10.7717/peerj.2551#supplemental-information.

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
