# Peer review of "Environmental and spatial drivers of taxonomic, functional, and phylogenetic characteristics of bat communities in human-modified landscapes"

_PeerJ, doi:10.7717/peerj.2551_

## Round 0.1 · original submission · Major Revisions

In particular, please focus on the issues raised by reviewer 2 under "validity of the findings". These issues must be clarified and addressed in the manuscript. With regard to Reviewer 1's concerns, please play special attention to style and use of language and minimise jargon (or at least explain it well) so a broader audience can be reached.

Reviewer 1 ·

Basic reporting

No comments

Experimental design

No comments

Validity of the findings

No comments

Additional comments

In this ms the authors used variation partitioning to explore the relative effects of spatial and environmental processes on taxonomic, functional, and phylogenetic structure of bat assemblages. The study examined a bat community in N Costa Rica that was sampled by mistnetting. The main conclusion is that landscape has a significant influence on assemblage structure, accounting for much of the variation in phylogenetic or functional dispersion. Although I am a bat ecologist, I have not a strong background on this kind of analysis, so my assessment may well be biased by my limited knowledge of this specific subject. As far as I can appreciate from the ms, the methodology employed appears rigorous and the results sound and discussed appropriately. My main concern is the overly technical narrative adopted, which I am afraid many in the journal’s readership will find quite difficult to fully comprehend, making the results themselves difficult to judge. This is all the more important since Peer J is a generalist journal tackling a very broad audience. Overall, my main suggestion is that the authors re-examine carefully the text making it simpler and easier to read, paying special attention to “translating” technical jargon into clear information when possible. Clarity is the first quality for any scientific articles: as authors, our main objective should be to convey an effective message. From this perspective I am convinced that this ms may be improved further and will need major revisions before it reads sufficiently well to deserve publication.

Reviewer 2 ·

Basic reporting

No comments

Experimental design

Please include in the methods a section where you state whether abundancnes were considered in all dimensions of biodiversity, or was it only considered in taxonomic structure?

L192- 195. Did the authors look into variable selection for funcional traits? It would be useful just to make sure there is no redundancy amongst those variables, and thus inflating type I error.

L208-209 - please explain in more detail how the environmental gradient may affect particular niche axes differently

L212-213 - Is weighting each niche axes equally is the best approach? is there a weight them based on importance? maybe PCA or variable selection?

Validity of the findings

I m a bit weary of the findings for phylogenetic and functional structure. Particularly of most variation being explained by the shared fraction. Can´t that reflect just the inherent spatial autocorrelation present in environmental gradients? Maybe address this in the discussion

L358-365 - it is not clear to me why do rare species drive patterns in functional and phylogenetic composition but for taxonomic structure its the abundant species?

L391-394 - I wonder if the differences amongst dimensions of biodiversity is related to the fact that taxonomic structure is being represented as a community structure, but phylogenetic and functional is represented as dispersion? Maybe address this issue in the discussion

Additional comments

Good introduction. Very complete and gets up to date with what the state of the knowledge in community assembly is related to using different dimensions of biodiversity.
The methods and data used are solid and there is no concerns over their validity to address the questions posed in the introduction.
Results are clear and figures are a good representation of these results. There a re a few concerns regarding the differences in data structure for taxonomic, funcional and phylogenetic dimensions that I recommend to be addressed in the discussion.

---

## Round 0.2 · accepted · Accept

It is clear that you have made a significant effort to clarify the complexity, and I am satisfied with your response to the methodological questions raised.